# Multi-Physics Analysis for Rubber-Cement Applications in Building and Architectural Fields: A Preliminary Analysis

**Marco Valente [1,2,\*], Matteo Sambucci [1,2], Abbas Sibai [1] and Ettore Musacchi [3]**

1   Department of Chemical and Material Engineering, Sapienza University of Rome, 00184 Rome, Italy; matteo.sambucci@uniroma1.it (M.S.); abbas.sibai@uniroma1.it (A.S.)
2   INSTM Reference Laboratory for Engineering of Surface Treatments, Department of Chemical and Material Engineering, Sapienza University of Rome, 00184 Rome, Italy
3   European Tyre Recycling Association (ETRA), 1040 Brussels, Belgium; ettore_musacchi@yahoo.it
\*   Correspondence: marco.valente@uniroma1.it; Tel.: +39-06-4458-5582

**Abstract:** Generally, in most countries, there are no strict regulations regarding tire disposal. Hence, tires end up thrown in seas and lands as well as being burnt, harming the living beings, and are therefore considered a very dangerous pollution source for the environment. Over the past few years, several researchers have worked on incorporating shredded/powdered rubber tires into cement-based material. This strategy shows a dual functionality: Economic–environmental benefits and technological functionalization of the building material. Rubber-modified cement materials show interesting engineering and architectural properties due to the physical-chemical nature of the tire rubber aggregates. However, the abovementioned performances are affected by type, size, and content of polymer particles used in the cement-based mixtures production. Whereas an increase in the rubber content in the cement mix will negatively affect the mechanical properties of the material as a decrease in its compression strength. This aspect is crucial for the use of the material in building applications, where proper structural integrity must be guaranteed. In this context, the development of innovative manufacturing technologies and the use of multi-physics simulation software represent useful approaches for the study of shapes and geometries designed to maximize the technological properties of the material. After an overview on the performances of 3D printable rubber-cement mixtures developed in our research laboratory, a preliminary experimental Finite Element Method (FEM) analysis will be described. The modeling work aims to highlight how the topology optimization allows maximizing of the physical-mechanical performances of a standard rubber-cement component for building-architectural applications.

**Keywords:** tire crumb rubber; rubber-cement mortars; additive manufacturing; FEM-based mechanical analysis; structural optimization; design

## 1. Introduction

Recycling processes, applied to the construction industry, aim to save the environment by reducing the need for extracting, refining, and processing mineral aggregates, as well as reducing wastes that have a negative impact as harmful chemicals and greenhouse gasses. In this context, the role of material engineering is to recycle theses wastes and ensure that these new materials are functional again for industries to produce new products and cementitious compounds for a cheaper price and with a lower environmental impact [1].

Waste tires are one of the main social and environmental issues worldwide. With the increase in the automotive industry, large amounts of waste tires cumulate in the world each year. Specifically,

an annual production of nearly 1 billion tires is estimated. Over 50% of these are discarded in landfills or stored in underground sites, without any treatment. Due to their poor biodegradability and impermeable nature, tires store water for a longer period, providing breeding habitats for the pests. This effect inevitably promotes soil pollution and the spread of many diseases [2,3]. Burning is another method for tire disposal, but the high temperatures involved in the thermal treatment imply the uncontrolled release of toxic gas and benzene compounds, potentially harmful to living beings and atmosphere [2].

As a promising solution to the tire disposal problem, the strategy based on the incorporation of rubber particles (produced by shredding and commutating used tires) in cementitious matrices has gained considerable attraction from many researchers working in the civil engineering and building sectors [3–5]. According to the previous works, the physical and viscoelastic properties of the recycled tire rubber aggregates confer to concrete material interesting engineering properties: Elasticity, lightweight, vibration damping, durability, sound, and thermal insulation [6,7]. However, the mechanical strength of rubberized material decreases significantly as the rubber content increase. This limits the incorporation of a large amount of polymer aggregates in the cement matrix, and hinders the use of these mixtures in several structural applications [8].

The mechanical weakness of the rubber-cement mixtures is the starting point of this research work. The main topic of this work is to combine the skills of materials engineering with the possibilities offered by the architectural design in order to maximize the physical–mechanical behavior through topological optimization approach. This means studying and testing functional shapes and geometries to be scaled in the manufacturing of rubber-cement bricks or building components. As highlighted by several research works, the type of perforated structure of brick can significantly influence its structural, thermal, and acoustic efficiency [9,10]. Thanks to the progress of the technical drawing and FEM (Finite Element Method) analysis software, it will be possible to perform rigorous studies on the physical–mechanical behavior offered by different morphology design. In this case, the aim of structural modeling is to select the shape of the inner holes that maximizes the mechanical performance of the component, also improving its lightness and thermo-acoustic efficiency.

The first part of the paper presents the mixes design and the experimental physical–mechanical characterization of several tire rubber-cement mortars obtained by total volume replacement of the sand with a different grain size of crumb rubber (0–1 mm rubber powder and 2–4 mm rubber granules). The innovative aspect of these cementitious mixtures concerns the possibility of using them in Additive Manufacturing (AM) processes. Indeed, the mix proportions were optimized to obtain rheology and wet properties suitable for the extrusion-based printing process. Such technology promotes potential benefits in the construction field (cost-effective, high efficiency, reduction in labor for safety reasons, environmentally friendly, design flexibility) and is therefore well suited to the automated production of complex structures that would often be difficult to produce using traditional manufacturing methods [11–13].

Specifically, three types of printable mixtures have been developed and investigated:

- Control mix: Standard cement mortar (100% sand) used as the basis for the preparation of the rubberized mixes and for comparison purposes.
- P100 mix: Rubberized cement mortar obtained by total replacement of the sand with rubber powder.
- P-G50/50 mix: Rubberized cement mortar obtained by total replacement of the sand with an equal proportion of rubber powder and rubber granules.

The goal of this investigation is to evaluate the effect of the rubber aggregates' size on the structural and performance of the material and to select the type of rubberized mixture with less mechanical loss than the standard mix. The "best" formulation was used as a benchmark rubber-cement mix for the computational analysis described below.

To complete the discussion on the printable rubber-cement compounds, FEM-based mechanical analysis is presented. This study is an advanced approach to enhance the properties of building materials and to investigate their possible technological applicability. In this regard, the proposal is to exploit the design flexibility, related to AM processes, to develop a rubber-cement brick with an engineering functional design. As a preliminary research stage, the effect of different inner morphologies on the mechanical performances was evaluated. Three types of brick designs were tested and compared with a reference component (unperforated brick): Two conventional geometries (round and square holes) and one unconventional inner structure (hexagonal holes). Referring to Munoz Guzman et al.'s research work [10], a honeycomb design would result in a relevant improvement in mechanical strength and thermo-acoustic damping compared to conventional cavities. The numerical simulation was performed using the commercial finite element software COMSOL Multiphysics v5.4. By the Structural Mechanics Module, a uniaxial compression test was simulated to reproduce the load conditions reported in ASTMC 67-03a standard [14]. The output FEM results were stress–strain curves, from which the mechanical properties (compressive strength and Young's modulus) were extrapolated for each type of brick prototype tested. The possibility of using the rubber-cement compounds in digital manufacturing methods implies a relevant versatility from design and architectural freedom points of view. In this regard, a concept of shape-optimized hollow brick with novel easy-to-assemble structure is also presented.

## 2. Materials and Methods

### 2.1. Materials, Print Process, and Samples Preparation

In this research, two different granular samples of recycled tire crumb rubber (ETRA, European Tyre Recycling Association) were used: Rubber powder (0–1 mm size) and rubber granules (2–4 mm size). The average density of the waste rubber particles, evaluated by Micromeritics AccuPyc 1330 He-pycnometer, is 1202 kg/m$^3$. An image of the crumb rubber used in this work is shown in Figure 1.

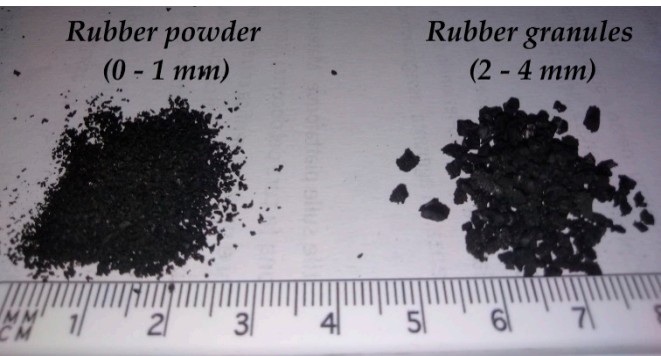

**Figure 1.** Crumb rubber used in this work.

To produce the rubberized mixtures, the polymer aggregates were incorporated in total volumetric replacement of the sand constituting a 3D printable standard cement mortar (labeled as Control). Control mix composition is described next (dosage for 1 m$^3$):

- Type I Portland cement: 800 kg
- Limestone sand (maximum size of 0.4 mm): 1100 kg
- Water: 300 kg
- Relation water to cement (w/c ratio): 0.375
- Silica fume: 120 kg
- Chemical additives (expansive agent, superfluidifying and reducer additives): 32 kg.

Starting with this composition, specific sand-rubber replacements were performed to obtain two sets of rubber-cement mortars (labeled as P100 and P-G50/50, respectively). The preparation of the investigated fresh mixtures was based on a three-step process: Premix, water addition, and mixing. In premix stage, dry constituents were weighed and mixed to ensure a homogeneous compound. Next, water was added to promote the cement hydration reaction. The proper dosage of water has been selected to obtain a specific w/c ratio and therefore achieve certain mechanical properties. Water-dry materials mixing was performed with an electric mixer drill for a time not exceeding 12 min. This time interval allowed the optimal effect of the superplasticizer additive and the obtainment of rheological properties compatible with the extrusion system used in this research.

To develop the printable rubber-cement compounds, all mix design parameters were kept constant except for the rubber aggregates constituents and the w/c ratio. In P100 mix, the sand was totally replaced with the finest rubber fraction (rubber powder). In P-G50/50 mix, the rubber content was divided equally between the powder and granules. The optimum fluidity and consistency, related both to the printing machine requirements and to the material extrudability [13], were analyzed through printability tests aimed to evaluate the print quality of the fresh compounds (easy-flowing, shape stability, buildability) and the structural features of the printed object (surface finish, inter-layer adhesion). These experimental analyses were performed at the Dept. of Materials, Environmental Sciences, and Urban Planning, Marche Polytechnic University (Italy), our technical supporter in the manufacturing processes, and owner of the printing apparatus. Additional details of printability test are available in previous work from our research activity [15]. Optimum mix designs of rubberized compounds are shown in Table 1.

**Table 1.** Mix proportions of rubberized compounds.

| Mix Design | P100 | P-G50/50 |
|---|---|---|
| Cement [kg/m$^3$] | 800 | 800 |
| Water [kg/m$^3$] | 260 | 250 |
| w/c ratio | 0.325 | 0.312 |
| Sand [kg/m$^3$] | 0 | 0 |
| Silica fume [kg/m$^3$] | 120 | 120 |
| Rubber pwd. [kg/m$^3$] | 300 | 150 |
| Rubber gran. [kg/m$^3$] | 0 | 160 |
| Additives [kg/m$^3$] | 32 | 32 |

For the printability test, a circular nozzle (Ø = 10 mm) was used to deposit the control and rubberized mortars layer by layer. The extruder is fixed on a 3-axis COMAU robotic arm, which performs a print path in accordance with the three-dimensional model data of the desired sample. The controls relating to robotic arm motion and the setting of the printing parameters were monitored through Cura® slicing software. The fresh material was placed in an Aluminum container connected to a peristaltic pump. The pumping allowed to convey the material to the printing nozzle through a PVC hose pipe. During the entire process, the pump flow rate and the printing speed (33 mm/s) were kept constant to avoid adverse effects on print quality. For each mixture, six-layer rectangular slabs (220 mm × 160 mm × 55 mm) were manufactured (Figure 2).

After the printing process, three types of specimens were taken out from the hardened slabs (Figure 2b) by sawing with a diamond blade:

- 48 mm × 42 mm × 22 mm blocks (Figure 3a)
- 50 mm-side cubes (Figure 3b)
- 40 mm × 40 mm × 230 mm beams (Figure 3c).

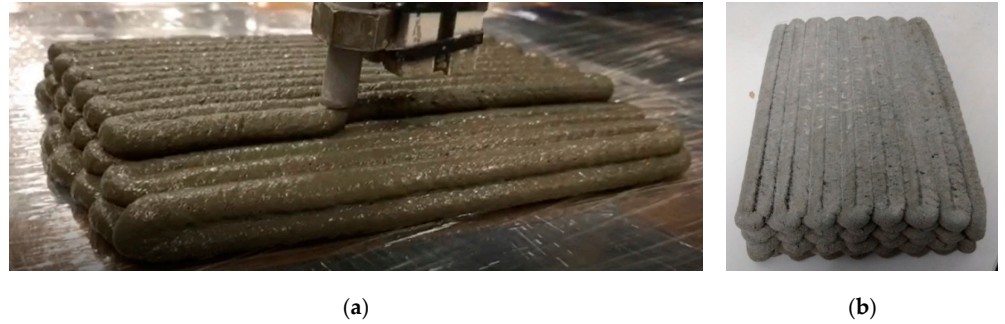

(**a**)　　　　　　　　　　　　　　　　(**b**)

**Figure 2.** Extrusion of fresh material (P100 mix) during the printability test (**a**) and six-layer 3D printed slab in hardened state (**b**).

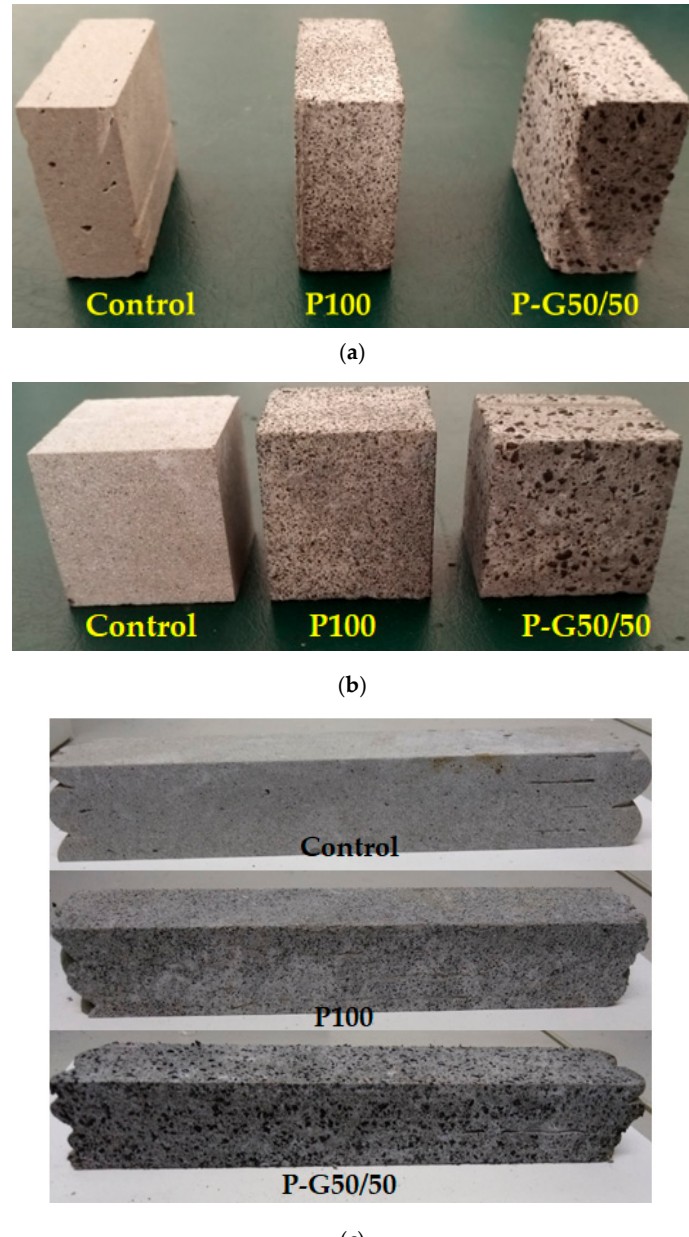

(**a**)

(**b**)

(**c**)

**Figure 3.** Specimens for physical and mechanical experimental testing: Blocks (**a**), cubes (**b**), and beams (**c**).

## 2.2. Testing Program

The physical and mechanical characterization included bulk density, porosity, compressive, and four-point flexural tests after 28 days of ambient curing. Four replicate samples were tested for each mix (control and rubberized compounds). Details of the experimental campaign are described below.

### 2.2.1. Physical Characterization: Bulk Density and Permeable Porosity

Bulk density was determined by geometric measurement and weighing on oven-dried blocks (110 °C for 24 h).

Based on ASTM C 1220 standard [16], the permeable porosity ($P_\%$) test was conducted. The blocks are weighed under three conditions: Oven-dried (110 °C for 2 h), vacuum-saturated (0.3 bar for 3 h in a desiccator connected to a vacuum pump), and distilled water-saturated (total soaked for 24 h). Then, the obtained weights ($W_{dry}$, $W_{sat}$, and $W_{wat}$, respectively) are used to calculate $P_\%$ based on the following equation (Equation (1)):

$$P_\% = \frac{W_{sat} - W_{dry}}{W_{sat} - W_{wat}} \tag{1}$$

### 2.2.2. Mechanical Characterization: Four-Point Flexural and Uni-Axial Compressive Tests

Four-point flexural tests (ASTM C 348) [17] were performed on a Zwick-Roell Z010 (10 kN) universal testing machine. Beams were tested to evaluate the effect of the rubber aggregates on Young's modulus and flexural strength of the material. The span length was 180 mm, and the loading rate was 1 mm/min. Figure 4a shows the details of the flexural test setup.

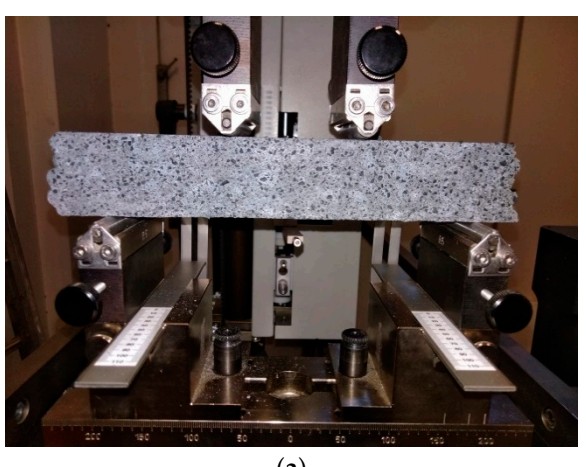
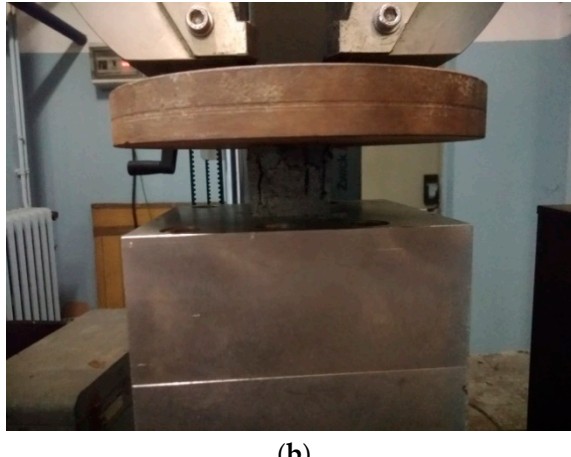

(**a**)　　　　　　　　　　　　　　　　　　　　　　　　　　(**b**)

**Figure 4.** Mechanical tests experimental setup: Four-point flexural test (**a**) and uni-axial compressive test (**b**).

Procedures for the uniaxial compressive test complied to ASTM C 109/109M standard [18]. The test was conducted on cube specimens subjected to an axial load applied by a Zwick-Roell Z150 (150 kN) machine with 1 mm/min rate (Figure 4b). For each mixture, the compressive strength result is the average of four specimens.

From experimental stress–strain curves, the compression toughness index (ASTM C 1018-97) [19] was also evaluated. This dimensionless parameter represents the relation between the plastic and elastic regions, then provides a measure of energy absorption capacity of the material. The index was computed by dividing the total area under the stress–strain curve up to a specific service level strain ($T_{sl}$) by the area under the same curve up to the peak stress ($T_p$). According to Khaloo et al.'s [20] work, 80% of peak stress was considered as service level. Figure 5 illustrates how each of the areas was determined from an experimental graph.

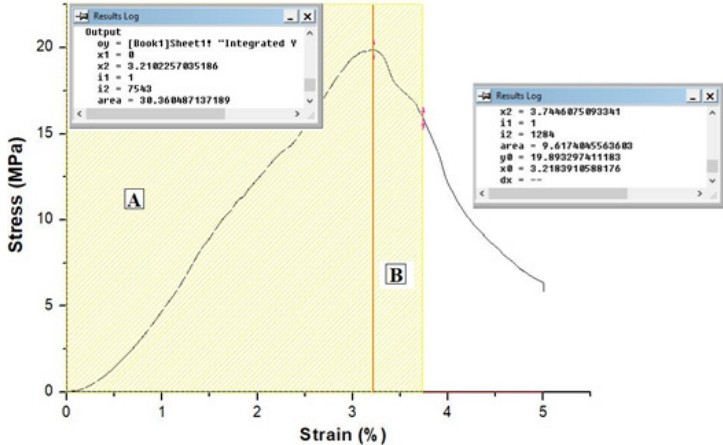

**Figure 5.** Evaluation of compression $T_i$ from stress–strain diagram.

Referring to Figure 5, the toughness index ($T_i$) is expressed as follows (Equation (2)).

$$T_i = \frac{T_{sl}}{T_p} = \frac{A + B}{B} \tag{2}$$

### 2.3. Finite Element Method (FEM)-Based Mechanical Analysis of Rubber-Cement Bricks

Modern numerical computation methods have wide applications for newly designed building components. FEM is a numerical procedure to solve several engineering problems such as acoustic, heat transfer, electromagnetism, and in this context, structural-mechanical analyses. In all cases, the influence of some parameters (material properties, mesh, boundary conditions, mathematical models) affects the numerical results [21,22]. For instance, the choice of an appropriate mathematical model is crucial and completely determines the insight into the actual physical problem that we can obtain by the analysis [23]. The general advantages of the FEM analysis are summarized below [24]:

- Easy modeling of complex shapes and irregular geometries
- Determine how critical factors could affect the investigated structure and why failures might occur
- Identify any design vulnerabilities and use the numerical results to develop a new design or perform topological optimization.

COMSOL Multiphysics v5.4 FEA software was used to evaluate the mechanical strength and elastic behavior for several brick designs. The assessment includes studying the effects of the hole architectures on the structural performances of modeled prototypes. For each building element, compressive strength and Young's modulus were calculated by simulating a compressive test under the experimental conditions reported in ASTMC 67-03a standard [14]. Modelling and simulation workflow followed in the numerical study is shown in Figure 6.

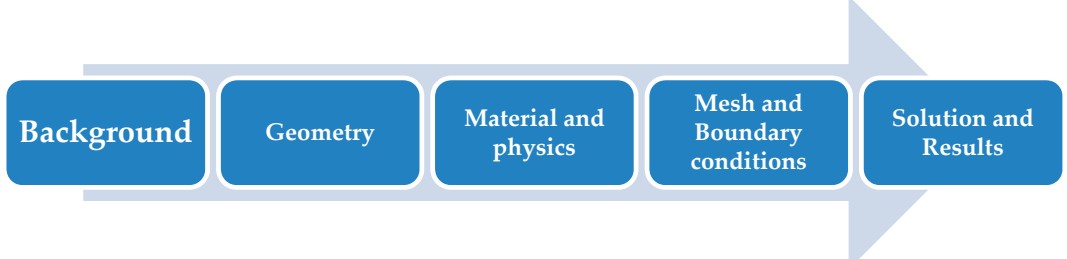

**Figure 6.** FEM analysis: Workflow.

The detailed description of each step of FEM analysis is discussed below.

2.3.1. Background

In this paragraph, some research works are reported showing theoretical/experimental studies on the morphological optimization of structural elements (hollow blocks, bricks, perforated sheets). As highlighted in these works, there is a close correlation between the design of the component and its mechanical, thermal, and acoustic properties. Based on the reported results, the brick concepts studied in this work by FEA were designed.

Khatam et al. [25] studied the effect of the geometric features of porous arrays (perforation pattern and holes shape) on the mechanical properties of perforated metal sheets used in the building sector as acoustic dampers or low strength/weight elements. Perforations reduce material's stiffness and improve the deformation capability. These act as additional discharge sites which reduce the concentration of stresses in the element (shadow effect). Then, the cracks are not concentrated in limited locations (as in the unperforated case) but spread in larger areas. Besides, comparing several types of perforation patterns, the hexagonal one provided the best mechanical strength performances.

Cabezas [26] highlighted a clear difference in mechanical behavior between horizontally perforated bricks and vertically perforated ones. Vertically oriented holes increase the compressive strength of the component by 5–10 times compared to the horizontal holes-based design.

Perforated bricks are also widely used because of their attractive properties when compared to traditional solid units (lightweight, thermal insulation, durability). In some works, the peculiarities of the perforated morphology on the thermal behavior were demonstrated. Through numerical and experimental studies, it has been verified that convection heat transfer is negligible in the perforations. This condition is necessary to increase the thermal resistance of the brick [27,28]. According to Laurenço et al.'s study [29], the thermal performance improves with the shape complexity of the hollow cells, once the path covered by the thermal wave is maximized. Baronio et al. [30] compared the deterioration of bricks, with and without cavities, due to saline crystallization. Perforated bricks exhibited greater durability, apparently because part of the crystallizing salt can accumulate in the holes, thus decreasing the total content of deteriorating agents in the brick matrix.

Munoz Guzman et al. [10] suggested a numerical study on "functional" internal designs aimed at increasing the mechanical and acoustic properties of building blocks. "Honeycomb" holes-based brick shows a compressive strength of 20% higher than conventional geometries (rectangular and triangular holes). Besides, computational acoustic analysis on hexagonal-shaped brick highlighted a noise damping 30% higher than that found in a traditional solid and rectangular-shaped bricks. The authors explained this evidence considering that in perforated bricks, the sound wave takes longer to travel through the component due to reflection phenomena against the external sides of the cavities. The higher the geometric complexity of the holes, the higher the number of wave reflections and therefore the dissipation of acoustic energy.

2.3.2. Geometry

Three types of vertically perforated bricks were modelled, all of them with the same dimension (250 mm × 150 mm × 40 mm). Hollow samples have the same number of holes (12) arranged in a double-hexagonal array. Circular (CS brick), square (SS brick), and "honeycomb" (HS brick) cavity shapes were investigated. The holes size was optimized to ensure the same perforation degree in each model (40%). In CS bricks, the cavity radius is 13 mm. In SS bricks, the square side is 33 mm. In HS bricks, the hexagon side in 20.5 mm. To evaluate the effect of the perforated structure on the mechanical behavior of the brick, the unperforated block (Std.) was also investigated. The geometries of models for solid and hollow bricks are shown in Figure 7.

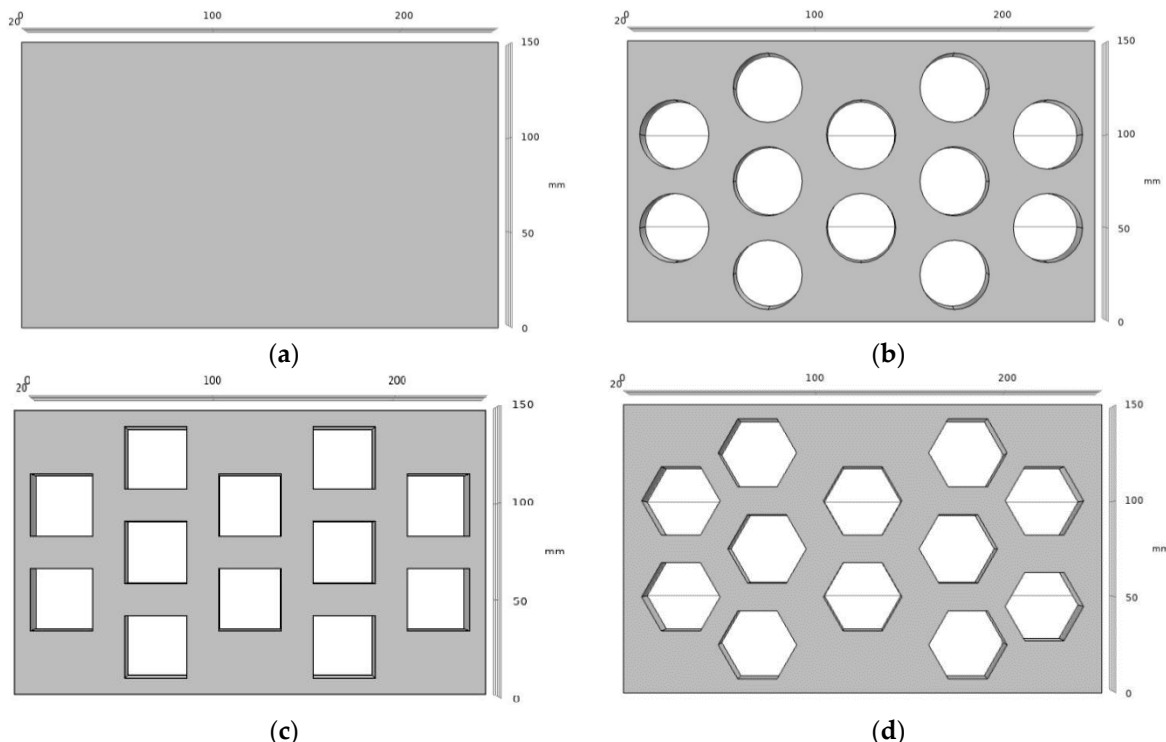

**Figure 7.** Top view of the brick models investigated in FEM analysis: Std. brick (**a**), circular (CS) brick (**b**), square (SS) brick (**c**) and honeycomb (HS) brick (**d**).

### 2.3.3. Material and Physics

3D FEM-based mechanical analysis was performed using the Structural Mechanics Module. It is dedicated to the investigation of mechanical structures that are subject to static or dynamic load. This interface allows us to perform stress and strain analyses in 2D and 3D coordinate systems for solids, shells, plates, beams, and membranes. A Time-dependent study was used to compute the mechanical behavior of the brick subject to a uniaxial compressive load. For this purpose, a static compression test was simulated on the bricks considering the load conditions in ASTMC 67-03a standard [14]. A 14 MPa compressive load was applied to the samples, recording 15 load-displacement values every 10 seconds. The simulation reaches convergence when the "computed" brick failure occurs.

For the accurate analysis of building components in macro-modelling approach, Willam-Warnke (WW) criterion [31] was adopted to predict the behavioral characteristic in uniaxial compression and failure modes of rubber-cement bricks. This model has been used in other research works and has proven to be a simple and precise model to predict the performance of concrete and mortar structures such as hollow blocks [32], composite beams [33], and masonry [34]. Implementation of the WW material model in COMSOL Multiphysics v5.4 requires three parameters that must be defined: Iniaxial compressive strength ($\sigma_c$), uniaxial tensile strength ($\sigma_t$), and biaxial compressive strength ($\sigma_{bc}$). Concerning the mechanical properties not experimentally evaluated ($\sigma_t$ and $\sigma_{bc}$), the following empirical relationships were considered:

- $\sigma_t$ of concrete material is 12 % of its $\sigma_c$ [35];
- $\sigma_{bc} \sim 1.2\sigma_c$ [36].

Table 2 summarizes the input material properties for the brick modelling.

**Table 2.** Input material properties for FEM-based mechanical analysis.

| Material Property | Symbol | Property Group | Evaluation |
|---|---|---|---|
| Density | $\rho$ | Basic | Experimental |
| Porosity | P | Basic | Experimental |
| Young's modulus | E | Basic | Experimental |
| Uniaxial compressive strength | $\sigma_c$ | Basic/WW criterion | Experimental |
| Uniaxial tensile strength | $\sigma_t$ | WW criterion | Empirical |
| Biaxial compressive strength | $\sigma_{bc}$ | WW criterion | Empirical |

### 2.3.4. Mesh and Boundary Conditions

Each model was assumed to be subjected to axial compressive load, as shown in Figure 8a. "Boundary load" condition was applied to the upper side of the brick. The load type is a pressure of 14 MPa. The bottom side was assumed fixed by setting "Prescribed displacement" equal to 0 in z directions. This assumption neglects the possible friction effect between the sample and the hypothetical support plane. A "free" boundary condition was selected for the other brick surfaces. A tetrahedral physics-controlled mesh (see Figure 8b) was used for the computational analysis. A "fine" element size was choice for mesh discretization in order to obtain a good compromise among results accuracy and computational power.

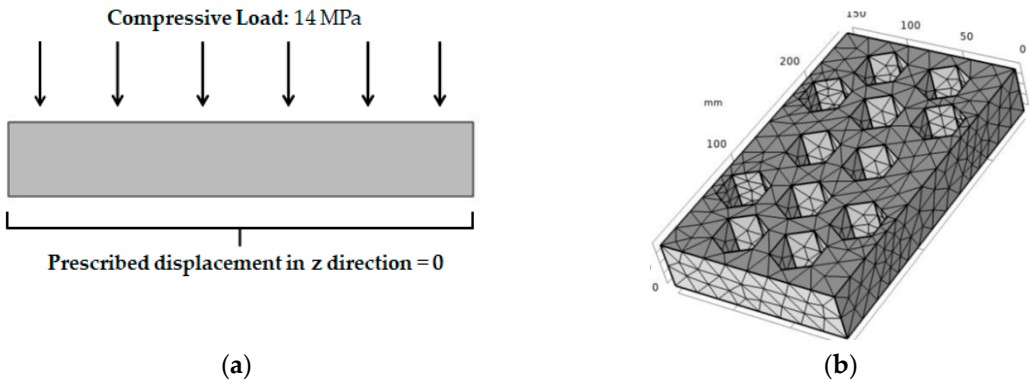

(a)  (b)

**Figure 8.** Applied load and boundary conditions (**a**) and mesh discretization of the model (**b**).

## 3. Results

### 3.1. Physical Characterization: Bulk Density and Permeable Porosity

Figure 9 shows the test results of bulk density and permeable porosity. The bulk density of mortars mixed with crumb rubber was found to decrease with the rubber content compared to control mix. Since the specific gravity of rubber aggregates is less than that of mineral ones, the sand-rubber replacement implied a reduction in dry unit weight [37,38]. The polymer finest fraction (rubber powder) had a more pronounced effect on material density. When comparing with control sample (1927 kg/m$^3$), the bulk density value of P100 mix was lower about 30%, while the percentage decrease of P-G50/50 was about 16%. The hydrophobic nature of rubber promotes the air entrapment during the mixing process of fresh material, resulting in a higher air content than the "neat" mortar. The amount of entrained air increases with decreased particle size. Rubber powder has a greater specific surface than rubber granules, so its capability to adsorb gas is most relevant [39]. Moreover, the unit weight discrepancy between the investigated rubber-cement mixes is also attributable to the overall amount of rubber incorporated into the material. In P100 mix, the total replacement of mineral aggregates with the fine polymeric fraction implies a greater number of polymer particles per unit volume than the "combined" P-G50/50 mix, resulting in a more relevant bulk density reduction. Bulk density values are typical for lightweight mortars and correspond to a RILEM class II [40] lightweight cementitious materials (oven-dry density less than 2000 kg/m$^3$). These class of cement

mixtures is generally used to reduce the dead weight of a structure and to improve the thermal insulation requirements in building applications [41].

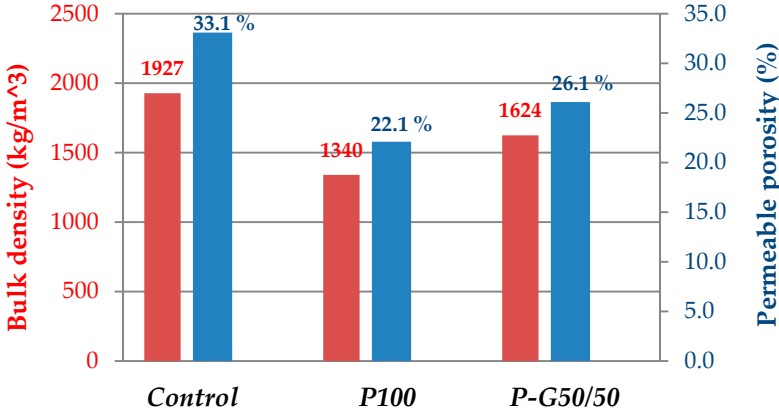

**Figure 9.** Physical characterization experimental results: Bulk density (red histogram) and permeable porosity (blue histogram).

Permeable porosity indicates the ease with which fluids and gas can penetrate and move through the cement material and it is, therefore, a good indicator of its durability. Besides, the voids fraction affects the mechanical strength of the hardened compound [42,43]. However, by promoting a certain porosity degree, the acoustic functionality of the material is enhanced. Several studies [44,45] highlighted how the pervious microstructure promotes the sound absorption properties of the cementitious materials.

Lower porosity is observed in the investigated rubberized mortars than the control mix. These experimental results can be explained for the reasons below [42,43]:

- As shown in Table 1, the mix design of the rubberized compounds requires less water amount than the control mixture. This aspect involves a lower capillary porosity in the hardened rubber-cement material
- Porous mineral aggregates were replaced with non-porous rubber particles. This factor reduces the rate of permeable voids in the material.

Comparing the rubber-modified mixtures, a slightly highest porosity degree is observable in P-G50/50 mix. This evidence can be explained by considering the effect of rubber aggregates surface morphology on the interface bond with the cement paste. As confirmed by Skripkiunas et al.'s research [46], the tires grinding degree affects the shapes and the surface texture of the tire particles. Coarse rubber fillers (up to 3 mm) show a very uniform surface. The shape and the surface texture become less regular and highly complex with smaller rubber particles. The rough texture in fine rubber aggregates ensures better interfacial bond strength and promotes the material compaction. According to the model proposed by Ghizdaveț et al. [42], during mixing, the fresh cement paste tends to penetrate in the rupture zone of fine rubber particle, developing an efficient anchoring mechanism when the material hardening occurs. On the other hand, the surface regularity in the coarse particles inhibits cohesion with the matrix, causing the generation of voids in the interfacial transition zone (ITZ) between the cement matrix and rubber particles. Figure 10 presents the optical micrographs, at 16× magnification, on investigated rubber-cement specimens. The purpose was to evaluate the adhesion properties of the rubber powder and rubber granules with the cement paste. The imaging analysis, performed with a Leica MS5 optical microscope on polished blocks, is in good agreement with the previous discussion. Fine rubber fraction (rubber powder) shows good adhesion with the cement paste (see Figure 10a). In P-G50/50 mix, the regular surface morphology of rubber granules hinders proper bonding with the matrix, involving the formation of cavities in the ITZ (see Figure 10b).

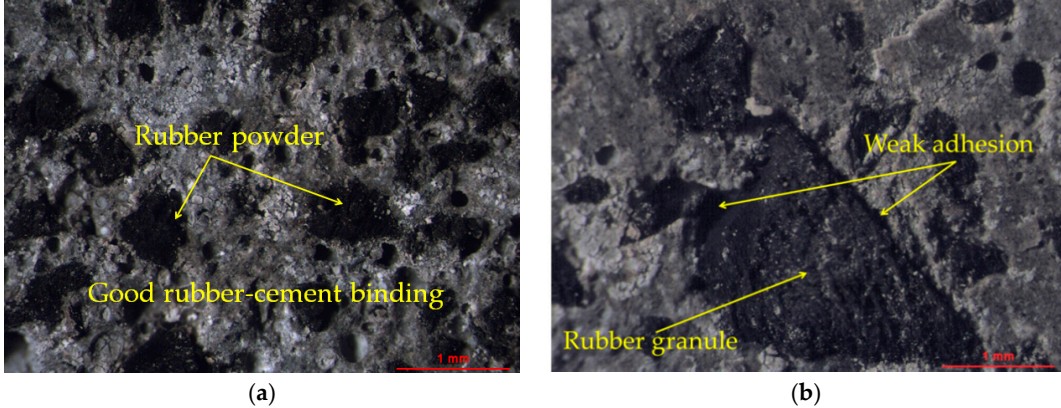

(**a**)                                                                                  (**b**)

**Figure 10.** Adhesion between rubber powder and cement paste in P100 mix (**a**) and between rubber granule and cement matrix in P-G50/50 mix (**b**).

*3.2. Mechanical Characterization: Four-Point Flexural and Uni-Axial Compressive Tests*

Flexural strength and Young's modulus of the investigated samples were obtained from four-point flexural test. The experimental results are presented in Figure 11.

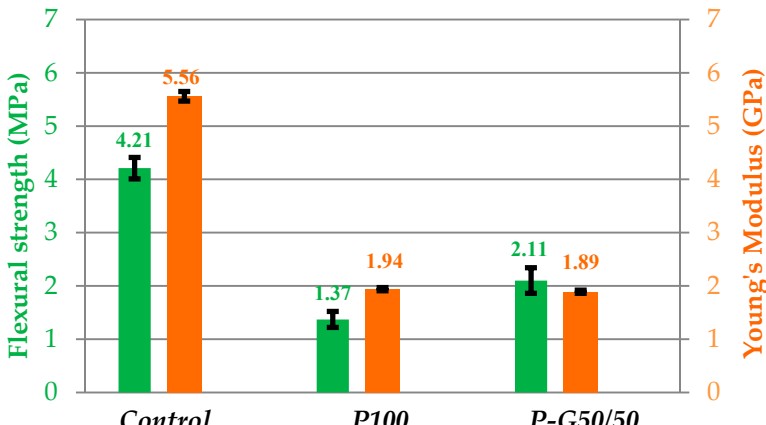

**Figure 11.** Four-point flexural test experimental results: Flexural strength (green histogram) and Young's modulus (orange histogram).

It was obtained that the rubber fillers reduced the flexural strength and the static Young's modulus compared to the control mix. The addition of the rubber particles changes the deformation capacity of the cementitious materials. The reduction in static elastic modulus may be explained by their low mechanical stiffness, which is much lower than the mineral aggregate [47,48]. The elastic behavior in both rubberized mixtures is very similar. This means that the rubber aggregate size does not significantly affect the material deformability. Considering the rubber-cement compounds as two-component composite materials (cement matrix + tire rubber particles), by the binary mixture rule, the elastic modulus will be related to the volumetric ratio and the Young's modulus of the rubber aggregate [49].

The total inclusion of rubber particles caused a remarkable loss in flexural strength for both types of rubberized specimens. Several reasons are behind the lower strength in rubber-based mortars [49,50]:

- Weak adhesion between rubber particles and cement paste. This implies an ineffective load transfer mechanism between cement matrix and rubber fillers.
- Lower mechanical strength of the polymer aggregates than the cement matrix. Such performance discrepancy makes rubber particles as structural weakness sources.

- The non-polar character of the rubber fillers increases the air content in the mixtures (then decrease in bulk density), promoting the worsening in mechanical strength.

Minor reduction in flexural strength was observed in P-G50/50 mix, where an optimal synergy among the coarse and fine rubber fractions occurs: Rubber powder promotes the material compaction, while rubber granules provide a better hindering effect to micro-cracks propagation [20] and reduce the tendency to trap air in the mixture.

The results of compressive test are illustrated in Figure 12.

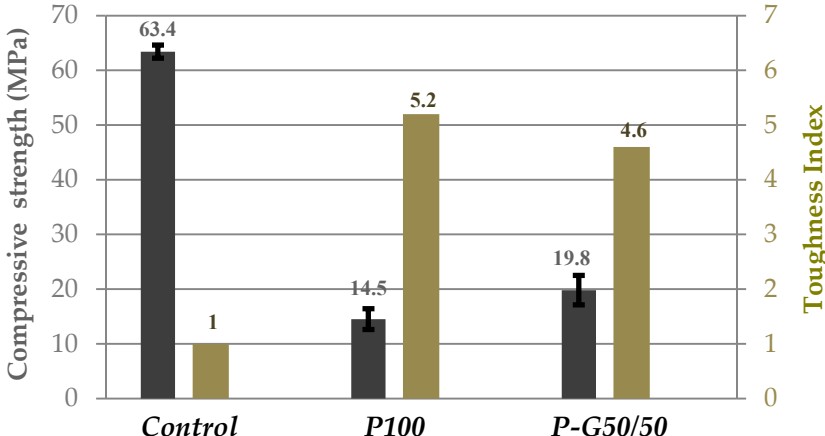

**Figure 12.** Uniaxial compressive test experimental results: Compressive strength (grey histogram) and toughness index (brown histogram).

Compressive strength data (Figure 12) follow the same trend as flexural strength. As mentioned above, the poor rubber-cement adhesion, the decrease in material density, and the mechanical mismatch among cement matrix and rubber fillers are the main causes of the mechanical strength reduction. Comparing to the control sample, P-G50/50 mix shows a lower mechanical loss (68% reduction) than P100 mix (77% reduction). Rubber powder-rubber granules synergy also provides a beneficial effect on compressive strength: There is a good balance between material compaction, trapped-air degree, and crack resistance.

The graph in Figure 12 also reports the average values of the $T_i$ for P100 and P-G50/50 specimens. Considering the brittle behavior of Control sample (with no post-peak load-carrying capacity), the $T_i$ value is equal to 1. The incorporation of tire rubber aggregates in cement compounds increases the material toughness. Thanks to their notable deformation properties, elastomeric particles act as structural deformation sites by absorbing deformation energy and reducing the cracks propagation in the matrix [51]. As confirmed by previous research work [20,52], a greater increase in material toughness occurs in the finest rubber aggregate-based mixture (P100) than the "hybrid" mix (P-G50/50). In P100 mix, the good rubber powder-cement paste interface adhesion ensures a more efficient and uniform load transfer, enhancing the elastic behavior of the fillers. The energy absorption, indicated by $T_i$ values, suggests several applications in which impact resistance or vibration damping are required [53].

### 3.3. FEM-Based Mechanical Analysis: Solution and Results

According to the experimental results discussed in the previous section, P-G50/50 mix properties were selected as input material data in FEM-based mechanical analysis (Table 3).

**Table 3.** P-G5/50 mix properties for FEM-based mechanical analysis.

| Material Property | Symbol | Value | Property Group | Evaluation |
|---|---|---|---|---|
| Density | $\rho$ | 1624 kg/m$^3$ | Basic | Experimental |
| Porosity | P | 0.261 | Basic | Experimental |
| Young's modulus | E | 1.89 GPa | Basic | Experimental |
| Uniaxial compressive strength | $\sigma_c$ | 19.8 MPa | Basic/WW criterion | Experimental |
| Uniaxial tensile strength | $\sigma_t$ | 2.38 MPa | WW criterion | Empirical |
| Biaxial compressive strength | $\sigma_{bc}$ | 23.8 MPa | WW criterion | Empirical |

Numerical load-displacement curves were analyzed to extrapolate the mechanical properties of each type of brick design investigated in this work. Table 4 shows the comparison of $\sigma_c$ and E values for the four proposed model. Comparing the mechanical strength of perforated bricks, CS, and HS designs offer the best result in terms of compressive strength. However, as previously reported, the "honeycomb" geometries have a remarkable thermal and acoustic functionality, therefore have more attractive requirements for building applications.

**Table 4.** FEM-based mechanical analysis: Numerical results.

| Brick Design | $\sigma_c$ (MPa) | Strength Reduction Rate | E (GPa) | Young's Modulus Reduction Rate |
|---|---|---|---|---|
| Std. brick | 21.2 | - | 1.87 | - |
| CS brick | 15.1 | 28.8% | 1.22 | 34.8% |
| SS brick | 13.4 | 36.8% | 1.22 | 34.8% |
| HS brick | 15.0 | 29.2% | 1.21 | 35.3% |

According to FEM analysis results, the rubber-cement mixture and the brick designs selected are all potentially suitable for the fabrication of load-bearing units. In this context, ASTM C90 [54] standards require minimum compressive strengths of 7.0 MPa and 11.7 MPa. As confirmed by the research mentioned in Section 2.3, perforated bricks show relevant stiffness reduction (~35%) compared to solid brick, due to a more favorable stresses distribution in the material. Cavity geometry doesn't significantly affect the elastic behavior, but has a noticeable effect on mechanical strength reduction. Small curvature radii of square cavities (SS brick) act as mechanical weakness elements, where remarkable stress concentrations occur, and cracks are easily triggered in the material. Conversely, when the cavity geometry tends to circular shape (wide curvature radii), improved performances in terms of compressive strength are observed. Figure 13 highlights the stress distribution comparison between Std. and HS brick.

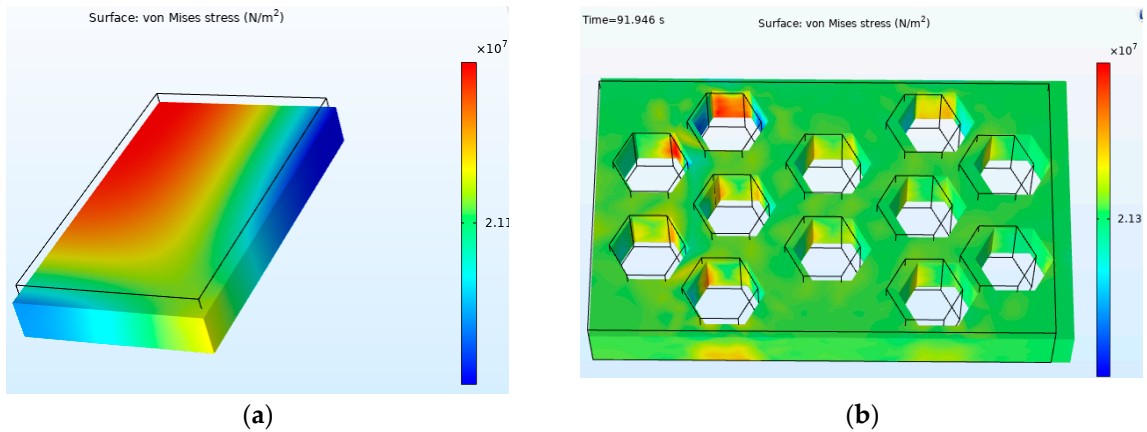

(**a**)          (**b**)

**Figure 13.** Stress distribution in Std. brick (**a**) and HS brick (**b**).

The relatively high flexibility of these modeled prototypes suggests their potential use in applications where mechanical/acoustic damping and ductility are primary requirements (flexible brick paving, lightweight partition walls, architectural units for insulation purposes).

## 4. Comparative Analysis of Tire Rubber-Cement Compounds Made by Casted and 3D Printing Methods

The use of granulated waste tires in concrete materials has been widely studied in the last twenty years as an eco-sustainable and functional solution in the civil and architectural fields. Besides saving natural resources and recycling end-of-life tires, rubber-cement mixtures could represent a class of building materials with improved energy efficiency. The enhancement of energy efficiency in the construction sector is one of the main challenges for a sustainable requalification of energy issues and a consequent favorable economic–environmental impact [55]. In this framework, the peculiarities of tire rubber-modified cementitious materials are attributable to the following aspects:

- Lightweight: The low density of rubberized mortars enables reduction dead-weight of structures and dimensions of building components (façade panels, slabs, block, non-structural elements). Weight reduction may reduce pre-cast elements transportation costs (less expensive handling and transporting equipment) as well as provide slender and spacious construction [56].
- Thermal insulation: The use of tire rubber in lightweight mortar allows the production of building materials with improved thermal efficiency. Pelisser et al. [57] evaluated the thermal performances of mortar panels, mainly for use in facades and signs. The addition of rubber aggregates gave the component a 15% higher thermal resistance than the standard material. This improved thermal insulation feature contributes to the achievement of energy efficiency, reducing operating costs through lower energy consumption.
- Vibroacoustic insulation: Lightweight rubber-cement mortars can be used for acoustic performance qualification in buildings for the impact and airborne noises. Concerning the sound transmission through a structure, rigid partitions that deform slightly before the vibration transmit the sound waves in a very short time. Elastomeric properties of tire rubber aggregates improve the material deformability, promoting the sound attenuation. Tutikian et al. [58] showed that the total replacement of mineral aggregates with polymeric aggregates results in an impact noise reduction of up to 15 dB.
- However, the possibility of using these compounds in AM methods represents an advanced technological upgrade. Preliminary comparison between the features of printable and casted rubberized mixes highlights some interesting effects brought by 3D printing, which can be key points in state-of-the-art context.
- Material rheology and physical-mechanical properties.

As mentioned in Section 2.1, the mix design of printable mixes must meet specific requirements: They must be extruded and support their shape without deformation or interruption. For this reason, the rheology of printable rubber-cement mixes is different from that of traditional ones, influencing the mechanical and microstructural properties of the hardened material. Table 5 shows a comprehensive comparison between the properties of some casted rubber-cement mixtures (deducted by literature research) and the 3D printable compounds proposed in this work. For a coherent comparative analysis, the literature data refer to rubberized mixes obtained by total mineral aggregates-tire rubber replacement.

**Table 5.** Rheological and physical-mechanical properties of casted and printed rubber-cement compounds.

| Sample Type | w/c Ratio | Density (kg/m$^3$) | Compressive Strength (MPa) |
|---|---|---|---|
| Casted [20] | 0.770 | <1300 | <1 |
| Casted [59] | 0.520 | 1300 | 1.5 |
| Casted [60] | 0.350 | 1321 | 7.1 |
| Printed (*P100*) | 0.325 | 1340 | 14.5 |
| Printed (*P-G50/50*) | 0.312 | 1624 | 19.8 |

3D printable compounds are characterized by a lower water dosage than casted samples. In the layer-by-layer deposition process, it is crucial to minimize the water content by acting on the proper balance of superplasticizer additives, to obtain an optimal consistency of the printed filaments and avoid collapse phenomena. Moreover, a low w/c ratio minimizes the formation of capillary porosity and air bubbles inside the material promoting less significant density reductions and therefore better strength performances.

### 4.1. Sample Manufacturing Method and Material Homogeneity

As observed by Thomas and Gupta [2], the rubber-cement samples made by mold casting process are characterized by the inhomogeneous distribution of polymer fillers into the cement matrix. Due to their low specific weight and poor adhesion with the cement, there is a tendency for the tire particles to move upwards during the vibration, promoting a greater concentration of rubber in the upper layer of the molded samples. The non-uniformity of the material adversely affects the mechanical properties, as confirmed by the strength value in Table 5.

On the other hand, the extrusion-based AM process allows a more homogeneous dispersion and alignment of the rubber aggregates due to both the deposition technology (Figure 14a) and the rheology of the material, which is designed with the aim of ensuring rapid mixture hardening after deposition. Figure 14b shows the internal section of P-G50/50 printed slab, where it is possible to observe the uniform dispersion of tire rubber aggregates.

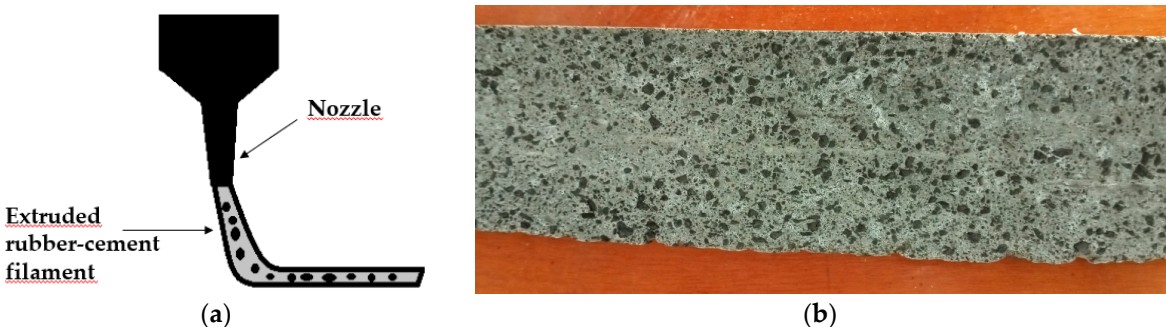

(**a**) (**b**)

**Figure 14.** Schematic illustration of the deposition process of printable rubber-cement mixes (**a**). The internal section of P-G50/50 slab (**b**).

Besides, the application of high pressure during the extrusion process ensures greater compaction and densification of the 3D printed parts compared to their casted counterpart [61], with consequent improvement in mechanical performance, as also confirmed by the comparative data in Table 5.

### 4.2. Constructability

The possibility of using rubber-cement mixtures in AM processes can lead to the formulation of a new technology that combines the engineering performance of the material with the design flexibility offered by digital manufacturing. In the design context, AM can be used to customize the shape of an architectural component in order to incentivize its performance in terms of architectural style, assembly, and maintenance. An example of architectural optimization is shown below (see Figure 15).

The prototype proposed by the authors is based on a Lego®-like shaped structure. Ease of assembly and the possibility to develop rounded architectural construction shapes could be the main advantages of this concept. In addition, the internal architecture of the brick can be designed to enhance the mechanical, thermal, and acoustic properties of the brick, as illustrated in the previous FEM analysis.

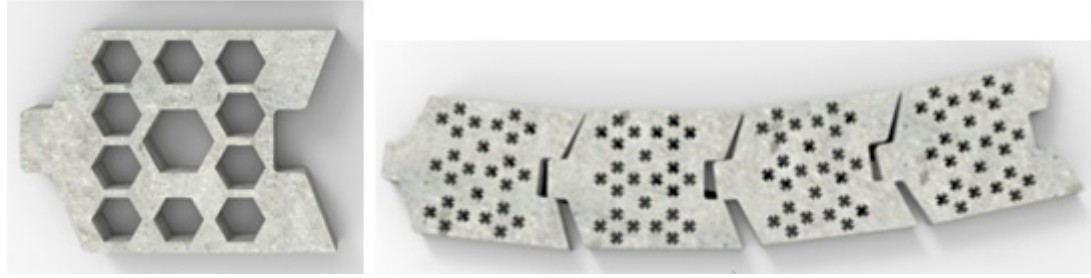

**Figure 15.** Brick concept with external "functional" morphology.

## 5. Discussion

### 5.1. Material Characterization

In this paper, the authors presented an innovative application of "rubber-concrete" technology. The possibility to modify a lightweight cement-based mixture, suitable for extrusion-based Additive Manufacturing, with recycled tire rubber aggregate (rubber powder and rubber granules) was investigated. Specifically, the rubber grain size effects on fresh compound printability properties and physical-mechanical performances were analyzed. The key findings from the experimental results are as follows:

- The total sand-rubber replacement does not alter the proper mixtures printability. Working on the w/c ratio, it was possible to obtain rheological properties suitable for AM process. Mix designs of rubberized compounds show a lower water amount than the "standard" fresh mortar. This evidence implies less capillary porosity in hardened rubber-cement material.
- Rubber powder-based mix (P100) is slightly less porous and more compact than the "hybrid" mix (50% rubber powder—50% rubber granules). Due to the complex surface texture, the fine rubber fraction develops an optimal interface bond with the cement matrix. Conversely, coarse polymer aggregates provide weaker adhesion with cement, promoting voids in the ITZ.
- The incorporation of tire rubber particles as aggregates in cement-based compounds decrease the bulk density in cement mixtures. Density reduction is lower in rubberized compound with polymer particles of combined sizes (P-G50/50 mix) than that of mix with 100% rubber powder. The presence of rubber granules in the mixture significantly reduces the rate of trapped air. On the other hand, the high specific surface of rubber powder promotes the tendency to absorb gas, with a consequent decrease in the unit weight of the material.
- In terms of mechanical properties, the rubberized samples showed a decrease in compressive strength, flexural strength, and Young's modulus compared to the "neat" material. The lowest strength reduction was related to P-G50/50 sample. This result can be attributed to the positive synergy between the two rubber fillers. Rubber powder ensures the material compaction and therefore an efficient load transfer among cement matrix and rubber aggregates. Rubber granules reduce the quantity of trapped air in the material and provide a hindering action to cracks propagation. Rubberized mortar was more deformable than Control sample, but the particles size didn't affect the elastic behavior.
- High $T_i$ values suggest that the rubber-modified cement materials have greater cracking and fracture resistance to static load than Control mix.

A possible implementing strategy could be focused on the chemical functionalization of tire rubber aggregates aimed at improving the rubber-cement adhesion and the performance of the hardened material in terms of mechanical strength and durability.

*5.2. FEM-Based Mechanical Analysis*

This piece of research presents an innovative approach in the modeling and structural optimization of rubber-cement mortars. FEM analysis is a very versatile tool for the study and physical-mechanical optimization of structures and building components. In this context, a numerical analysis was performed to investigate the effect of inner design on the mechanical properties of a rubber-cement brick. According to literature studies and the numerical results, it has been possible to deduce that the "honeycomb" perforated structure provides attractive performances in terms of mechanical flexibility, thermal insulation, and acoustic damping. The present analysis is a preliminary approach to this type of experimentation. Obviously, additional FEM-based multi-physics studies will be needed to evaluate further functional design, investigate a wider range of engineering requirements, and validate the numerical results with laboratory tests on real prototypes.

*5.3. Casted and Printed Rubber-Cement Compounds: Comparative Analysis*

The applicability of "rubber-concrete" technology to AM is an innovative topic in the construction and civil fields. Although the functional properties of tire rubber-modified cementitious materials are well known, the advantages that digital manufacturing can bring to the material are still to be explored. However, a preliminary comparative analysis between casted and 3D printed materials was presented in this work. The rheology of the printable materials combined with the deposition method would make it possible to obtain more homogeneous and compact samples with higher mechanical properties than traditional ones. Furthermore, the design flexibility of 3D printing would allow more sophisticated structural optimization studies aimed at improving the engineering and architectural performances of potential rubber-cement building elements.

## 6. Conclusions

The paper proposes an extensive research activity on novel rubber-cement mortars suitable for AM technologies.

First, the physical–mechanical properties of printable rubberized mixtures, obtained by total replacement of the sand with two tire crumb rubber fractions, were studied. The presence of the double polymer grain size in the mixture ensures lower loss in mechanical strength compared to the neat cementitious material, maintaining satisfactory deformability, toughness, and porosity properties. Besides, digital manufacturing provides mixture rheology and a deposition process which ensures a more compact, homogeneous material with greater mechanical strength properties than casted samples. However future research will be aimed at optimizing these cementitious compounds in terms of increasing rubber-cement compatibility.

A potential building application of these mixtures as functional hollow bricks was also reported. FEM-based mechanical analysis was performed to evaluate the mechanical performances of different internal designs. "Honeycomb"-shaped holes would seem to provide the best functionality in terms of mechanical properties (as confirmed by FEM results) and thermo-acoustic performances (as deduced from literature studies). Rubberized compounds–AM compatibility will allow further studies to performed on engineering attractive geometries, to obtain "bifunctional" elements, where a synergistic effect between material properties and topological functionality occurs.

**Author Contributions:** Conceptualization, M.V. and M.S.; methodology, M.V., M.S. and A.S.; software, M.S. and A.S.; validation, M.V., M.S. and A.S.; formal analysis, M.V. and M.S.; investigation, M.S. and A.S.; resources, M.V. and E.M.; data curation, M.S. and A.S.; writing—original draft preparation, M.S. and A.S.; writing—review and editing, M.V. and E.M.; visualization, M.V. and E.M.; supervision, M.V.; project administration, M.V.; funding acquisition, M.V. and M.S. All authors have read and agreed to the published version of the manuscript.

**Funding:** This research was performed thanks to the Sapienza University direct financing for PhD student Matteo Sambucci called "Avvio alla Ricerca". Title is: "Study and optimization of rubber-concrete additives with recycled rubber that can be used through additive manufacturing: Optimization of thermo-acoustic, rheological and mechanical properties."

**Acknowledgments:** The authors express their sincere gratitude to Valeria Corinaldesi and Eng. Glauco Merlonetti (Marche Polytechnic University) for their support in the additive manufacturing process. Besides, the authors would like to thank Rocco Crescenzi (Sapienza University of Rome) for the concession of COMSOL workstation.

**Conflicts of Interest:** The authors declare no conflict of interest.

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
