# Peer review of "Multi-Physics Analysis for Rubber-Cement Applications in Building and Architectural Fields: A Preliminary Analysis"

_sustainability, doi:10.3390/su12155993_

Round 1
Reviewer 1 Report
Dear authors,
This topic is interesting, However, this paper is very poorly organized. Three irrelevant sections are present without significant correlation. Seems like authors simply put three reports together. This is a wrong structure for a scientific paper. Please make more effort on scientific writing and make it seriously.
Overall the discussions are generic and some basic mistakes are present. For example, in Figure 6, why does P-G50/50 have a higher density with a higher porosity compared to P100? This doesn't make any sense.
Author Response
Cover letter for Reviewer 1
Dear authors,
This topic is interesting,
However, this paper is very poorly organized. Three irrelevant sections are present without significant correlation. Seems like authors simply put three reports together. This is a wrong structure for a scientific paper. Please make more effort on scientific writing and make it seriously.
We have revised the manuscript and edited its structure in order to ensure a better connection between the three sections proposed. We have partially removed the third section and added a comparative analysis between printable and traditional rubber-cement mixtures, considering it more useful for the scientific community. All changes and implementations of the manuscript have been highlighted through the Track Changes function.
Overall the discussions are generic and some basic mistakes are present. For example, in Figure 6, why does P-G50/50 have a higher density with a higher porosity compared to P100? This doesn't make any sense
In line 339, we added a piece of text aimed at clarifying the discrepancy between porosity and density values in rubberized mixtures. There are two competitive factors that affect the density of the rubber-cement mixtures: the porosity degree and the amount of tire rubber incorporated in the material. In rubber powder-based mix (P100), the amount of rubber per unit volume is greater than P-G50/50 mix and this implies a more significant reduction in unit weight. In our opinion the other discussions of the experimental results would also seem consistent with the literature data.
We have carefully reviewed and corrected the English language and style

Reviewer 2 Report
Please specify how many print layers are present in sawed off hardened slab for printability test.
Section 2.3: All the discussion provided compares the properties between control and rubber containing samples adequately. However, there is a lot of literature on rubber embedded cement composites which were not printed, but nevertheless provide a basis of material behavior. It would be good to provide material properties baseline in comparison to unprinted rubber cement composites to compare the printed sample performance in state-of-the-art context.
Author Response
Cover letter for Reviewer 2
Please specify how many print layers are present in sawed off hardened slab for printability test.
In line 163, the number of printed layers (6 printed layers) has been reported
Section 2.3: All the discussion provided compares the properties between control and rubber containing samples adequately. However, there is a lot of literature on rubber embedded cement composites which were not printed, but nevertheless provide a basis of material behavior. It would be good to provide material properties baseline in comparison to unprinted rubber cement composites to compare the printed sample performance in state-of-the-art context.
We sincerely thank the Reviewer for such a valuable suggestion. We implemented the paper by adding a section (Section 4, line 465) dedicated to a comparative analysis between traditional (unprinted) and printable tire rubber-cement compunds.
We have carefully reviewed and corrected the English language and style
Reviewer 3 Report
this is an interesting and well-organized work. this work can be published in the present form. thanks for sharing it with me.
Author Response
Cover letter for Reviewer 3
This is an interesting and well-organized work. this work can be published in the present form. thanks for sharing it with me.
We would like to thank the Reviewer for their valuable time and feedback